# Grading Criteria of *Anthurium* DUS Quantitative Characteristics by Multiple Comparison

**DOI:** 10.3390/plants12132417

**Published:** 2023-06-22

**Authors:** Yunxia Chu, Li Ren, Shan Deng, Shouguo Li, Yiying Zhang, Hairong Chen

**Affiliations:** 1Institute for Agri-food Standards and Testing Technology, Shanghai Academy of Agricultural Sciences, Shanghai 201403, China; chuyx@189.cn (Y.C.); zyy425zoey@163.com (Y.Z.); 2Shanghai Sub-Center for New Plant Variety Tests, Ministry of Agriculture and Rural Affairs, Shanghai 201415, China; lsgwjh@outlook.com

**Keywords:** grading criteria, *Anthurium*, multiple comparison

## Abstract

The determination of the grades and interval of quantitative characteristics is an important job while we draft new distinctness, uniformity and stability (DUS) test guidelines. Grading criteria should be adjusted because of the effect of year and site; it is also a key task to establish applicable criteria in the DUS test. Excellent criteria will improve the accuracy of the DUS evaluation. In this study, we analyzed the variability and distribution patterns of nine quantitative characteristics of 251 anthurium varieties. Three methods were used to establish the grade criteria: the two standard deviation methods, the two LSD_0.05_ methods and the multiple comparison method. The results showed that the coefficient of variation within varieties varied from 6.96% to 10.11%. The quantitative characteristics observed in this study did not follow a normal distribution, except spadix thickness at the middle and spathe size. In most characteristics, the standard deviations and LSD_0.05_ were similar, except for spathe size. The state interval set by multiple comparison methods for every characteristic was variable, and its mean was about 1.25 times that of the other two methods. The process of establishing grading criteria using the multiple comparison method was simpler, and the criteria were more accurate, with a lower error rate.

## 1. Introduction

The protection of new plant varieties is an important part of intellectual property protection and a key element of national intellectual property strategies [1]. Distinctness, uniformity, and stability (DUS) are the three essential criteria that a plant variety must meet to be eligible for Plant Breeder’s Rights (PBR) protection [2]. The International Union for the Protection of New Varieties of Plants (UPOV) is an intergovernmental organization based in Geneva, Switzerland, whose mission is to provide and promote an effective system of plant variety protection, with the aim of encouraging the development of new varieties of plants, for the benefit of society [3]. UPOV has developed a series of DUS guidance, including a general introduction to DUS and the associated series of documents specifying test guidelines procedures and 338 crop-specific test guidelines. In order to provide varieties to be tested and a variety description to be established, the range of expression of each characteristic in the Test Guidelines is divided into a number of states for the purpose of description, and the wording of each state is attributed a numerical ‘Note’ [4].

There are three types of DUS characteristics: qualitative characteristics, quantitative characteristics and pseudo-qualitative characteristics [5]. Quantitative characteristics are those where the expression covers the full range of variation from one extreme to the other [6]. The expression can be recorded on a one-dimensional, continuous or discrete, linear scale. The range of expression is divided into a number of states for the purpose of description (e.g., length of stem: very short (1), short (3), medium (5), long (7), very long (9)). The division seeks to provide, as far as is practical, an even distribution across the scale. It is the intention that the states and notes in the Test Guidelines are useful for the assessment of distinctness.

Test guidelines are the basis for DUS testing. The main problems for developing DUS Test Guidelines include the characteristics selecting, dividing expression states, and the selection of example varieties [7]. For the qualitative characteristics and pseudo-qualitative characteristics, the states are divided directly based on observation results, while for the quantitative characteristics, 5 scales, “1–9” scale, “1–5 “scale, “1–3” scale,”1–4” scale and “>9” scale were recommended in TGP/7 [8]. The suitable scale should be selected by the feature of the species. But there are only a few studies conducting research on dividing expression states of quantitative characteristics [9]. However, the traditional equidistant grading or empirical grading has certain limitations and often fails to accurately reflect the median and discrete degree of characteristic variation and the systematic position of the values taken at each level in the overall variation [10]. Suitable grading criteria for quantitative characteristics are an important guide for distinctness evaluation. Setting too many grades will lead to a high misjudgment rate. On the contrary, too few grades may affect the application enthusiasm because varieties need larger differences to be granted plant variety rights. It lacks a uniform scientific grading method. Two SD methods [11], two LSD_0.05_ methods [12] and the equal intervals method are often used to establish grading criteria. However, while SD or LSD_0.05_ is used, the minimum requirement for the multiplier is 2, but determining the exact multiplier factor would be difficult.

Whether directly using UPOV testing guidelines or developing national guidelines, the DUS exam, under the guidance of test guidelines, will perform before the registration of a new variety. Distinctness assessment of a new variety apparently looks easy, but actually, it is not so. Based on field and laboratory trials along with the most similar variety, a new variety is compared for all the characteristics which describe the variety according to the test guidelines. The new variety must be clearly distinguishable by one or more essential characteristics from any other variety whose existence is a matter of common knowledge at the time when the protection is applied to [13]. Although some statistical procedures such as COYD are used to make the comparison scientific and valid [14], note comparison method is more often applied in the DUS test. For quantitative characteristics, a difference between two notes often represents a clear difference. Varieties with the same note in the UPOV Test Guidelines for a given characteristic would not normally be considered to be clearly distinguishable with respect to that characteristic [15]. In test guidelines, many quantitative characteristics are recommended for using measurement methods, which means using a ruler, weighing scales, colorimeter, dates, counts, etc. Then, it is transferred to note according to the grading criteria [8]. As the quantitative traits, genetic control is high because the involved genes are numerous, with usually minor effects and very sensitive to the environment [16]. To increase the comparability of various descriptions from different years and sites, the grading standards need to be adjusted according to the expression of example varieties in the same trial; it is also a key task to establish applicable criteria in the DUS test.

*Anthurium* is an attractive and commercially popular ornamental plant used as a cut flower, flowering potted plant and landscape ornamental. Among the tropical foliage, the genus *Anthurium* excels in the ornamental market due to its rich diversity in shapes, beautiful leaves and durability [17]. The volume of *Anthurium* sales is ranked second in the world after orchids [18]. Anthurium breeding is gaining importance as it is a prominent place in the floral market. Classical and biotechnological methods are used in breeding [18]. Crossbreeding, especially the interspecific cross, contributed to a significant increase in anthurium varieties. Due to the high requirements of temperature, humidity, and light for anthurium, protected cultivation is commonly used for commercial breeding and production. Anthurium was introduced to China in the 1970s; it has developed rapidly in recent years and has become a major producer and seller of anthurium [19]. In China, the application for new variety protection of anthurium is relatively active, only less than *Chrysanthemum* and *Phalaenopsis* among the ornamental plants; there were 366 applications till August 2022. Although the “1–9” scale is determined in the national test guidelines, there is still much room for interval adjustment. Normally the anthurium DUS testing should be conducted for one growing period, so COYU and COYD are seldom used; note that comparison is the main method of distinctness assessment. Suitable grading criteria will help the tester to make an accurate determination of distinctness.

To explore the feasibility of using multiple comparison methods to establish grading criteria for quantitative traits, we analyzed the variability and distribution patterns of nine quantitative characteristics of 251 anthurium varieties and applied the multiple comparison methods to establish grading criteria for anthurium. This study was conducted to provide a new method to analyze the quantitative characteristics and set scientific grading criteria.

## 2. Results

### 2.1. Analysis of Variation in Quantitative Characteristics of Anthurium

Statistical analysis of quantitative characteristic data from 251 anthurium varieties revealed that the CV within variety ranged from 6.96% to 10.11%. Petiole length had the highest CV, while the lowest CV was observed in spadix thickness at the mid-point (Table 1). The low CVs indicated that there was relatively little variation within the anthurium varieties; it meant anthurium varieties have higher genetic uniformity and stability. Due to the application methods of anthurium being diverse, the varieties included both large varieties for cutting flowers and middle or micro varieties for potted ornamental. The measured values of quantitative traits were greatly varied, especially in petiole length; the maximum value is 6.5 times that of the minimum. In other quantitative traits, the maximum value is about three times the minimum value. This suggests a fine genetic diversity in the anthurium.

### 2.2. Test of Normality of Quantitative Characteristics

The absolute extreme difference is the larger value of positive difference and the absolute value of negative difference. Obviously, if the difference between the distribution of the sample population and the theoretical distribution is not obvious, then the absolute extreme difference should not be large; otherwise, the distribution of the sample population is quite different from the theoretical distribution. 

The K-S normality test showed spadix thickness in the middle, and spathe size conformed to a normal distribution, with sigma values greater than 0.05. Other quantitative characteristics did not conform to a normal distribution; their sigma values were less than 0.02 (Table 2). Among them, the values of leaf blade length, petiole length, peduncle length, spadix length and spadix thickness at the middle are positively skewed.

Therefore, the LSD_0.05_ method, probability grading method and other methods commonly used for quantitative characteristic grading maybe cannot work well for *Anthurium*.

### 2.3. Correlation Analysis Results of the Quantitative Characteristics

Correlation analysis showed that all quantitative characteristics of *Anthurium* were significant positive correlations (Table 3). The correlation coefficient varied between 0.484 and 0.893. The correlation coefficient of spathe size and petiole length was the smallest, only 0.484. The correlation coefficient of peduncle thickness, spathe size and spadix thickness in the middle with other traits was small, all less than 0.8. 

### 2.4. Quantitative Characteristic Grading Criteria of Anthurium set by Three Methods

The intervals of most characteristics obtained by the two SD methods and two LSD_0.05_ methods were equal or similar; the difference is less than 7%, except for spathe size (Table 4). The state interval set by multiple comparison method of every characteristic was variable, changed in the amplitude of 4.2–17.9%, to ensure that there is a significant difference between varieties with the two notes difference; the maximum of these values can be used as the grading interval. And its mean was about 1.25 times that of the other two methods. It deduced that in *Anthurium*, 2.5 × SD or 2.5 × LSD_0.05_ was appropriate as a state interval while the SD method or LSD_0.05_ was used.

All varieties were given notes based on the grading criteria established by the three methods, respectively, and the frequencies of each note were counted. The results showed that leaf blade length, petiole length and peduncle length have similar distribution patterns, with the first five levels containing more than 80% of the varieties and the decline rate of the last four levels slowing down to a normal distribution (Figure 1b,d,e). The frequencies of the other six characteristics were similar to the normal distribution graph, showing a pattern of high in the middle and low at the ends (Figure 1a,c,f–i), and using the criteria established by the multiple comparison method, the distribution frequency of each state was more inclined towards a normal distribution.

### 2.5. Effectiveness of Two Grading Criteria Set by Two SD Method and Multiple Comparison Method

The SD and LSD_0.05_ of most characteristics were similar, so only the two SD method was compared with the multiple comparison method here.

The distribution range at the middle class of the two SD methods was median ± SD, which would work well for traits that conform to a normal distribution, but if not enough resources were collected, or if disturbances in the breeding process led to a skewed distribution of trait, such as most of the quantitative traits in this paper, it would result in the need to shift the grading criteria left or right to establish relatively reasonable grading criteria, yet the shift distance lacks basis. In this study, only the spathe size and spadix length were not shifted; median ± SD was exactly the graded range of intermediate state (5). Among them, five traits, plant size, leaf blade width, peduncle length, peduncle thickness, and spadix thickness at the middle were shifted to the left by two notes; it meant that median ± SD was the range of note 3, while leaf blade length and petiole length were shifted to the left by 1 note (Table 5).

Grading criteria 1 and 2 were obtained according to the SD method and multiple comparison method, respectively. According to the two criteria, the measurements of all varieties were converted into notes, and the pairs with diverse difference value (D-value), the same note (0 D-value), adjacent notes (1 D-value), and two notes differences (2 D-value) were compared by the multiple comparison software, and the proportion of pairs accessed to be different was calculated (Table 6). Results showed that all pairs with the same note based on criteria 2 were accessed having no difference, while 23.84% of pairs were different based on criteria 1. Only 0.34% of pairs with 2 D-value of criteria 2 were not different, while 27.22% of pairs of criteria 1 had no difference. This meant that if two notes’ differences in the measured quantitative traits were considered to be significantly different according to the general rules of UPOV, the error rate was as high as 27.22%.

## 3. Conclusions and Discussion

The quantitative characteristics of anthurium observed in this study did not follow a normal distribution, except spadix thickness at the middle and spathe size. The variation coefficient within varieties varied from 6.96% to 10.11%. The grading results showed that in most characteristics, the standard deviations and LSD_0.05_ were similar, except spathe size. Grading by the multiple comparison method was simpler, and the criteria were more accurate, with a lower error rate. 

It is generally accepted that in the natural state, continuous or intermittent variables of biological phenomena conform to a normal distribution [20]. Many statistical procedures, such as correlation, regression, *t*-tests, and ANOVA, namely parametric tests, are based on the normal distribution of data [21]. However, the majority of characteristics observed in this article did not conform to a normal distribution; this result was consistent with the results of previous studies [22]. Research results and my statistical analysis showed that there were significant positive correlations among most quantitative characteristics in *Anthurium* [23]. Selection can change not only the means of quantitative traits but also their distributions, including variance and skew [24]. It is possible that breeding preferences were the main reason for most quantitative characteristics of *Anthurium* did not conform to normal distribution. Anthurium is cultivated primarily for its showy flowers and glossy leaves. The important horticultural features of the flower are its color, size, texture, shape and showiness of the spathe, spadix length, and peduncle length [25]. Breeders typically prefer varieties with long peduncles because a longer peduncle of potted varieties is usually associated with higher ornamental value, as the spathes are higher than the leaves, while in cut flower varieties, a longer peduncle usually has a higher market value. As anthurium evolved as an understory species in tropical forests [26], fewer leaves with larger leaf sizes may have been an adaptive feature. But during the sympodial phase, one flower is produced from each leaf axil [27]; fewer leaves also mean fewer flowers, which is a shortcoming not only for pot flowers but also for cut flowers. To improve the number of flowers, varieties with shorter or narrower leaf blades are preferred. 

A difference of two notes is appropriate if the comparison between two varieties is performed at the level of notes. If the difference is only one note, both varieties could be very close to the same borderline (e.g., the high end of note 6 and the low end of note 7), and the difference might not be clear. When comparing the measurement data, a difference smaller than two notes might represent a clear difference. To ensure the accuracy of distinctness assessment by note, appropriate grading criteria need to be taken into account first. The results of this paper showed that the criteria obtained by two SD methods would result in 27.22% of incorrect determinations, while only 0.34% by multiple comparison method. On the other hand, 23.84% variety pair with the same note by criteria obtained of two SD methods considered to be not clearly different was distinguishable if the statistic method was used. In the species with high breeding levels, such as rice and maize, despite the richness of genetic resources, reduced genetic base and the prevalence of only a small set of germplasm resources or landraces in the breeding process had been the general approach [28]. This breeding process has led to severe homogenization among the varieties and the very high similarity of morphological traits. Varieties with the same note of these species are often evaluated as distinct if it is differently deduced by a T-test or another statistical method. This means that if the grading criteria are not appropriate, even the varieties with the same note need to be statistically analyzed, which will greatly increase the computational work. The note obtained by measurement will lose its function. 

In research work, we may often have to determine whether differences exist among the means of three or more groups. The only way to answer this question is to apply the ‘multiple comparison test’ (MCT), which will clarify the differences between particular pairs of experimental groups. The earliest example of a multiple comparison procedure could be found in 1929 [29]. In DUS testing, it is a common occurrence that a candidate variety needs to be compared with similar wide varieties; MCT can help testers make a judgment. But there is no report about using MCT to group a large number of treatments. The variation of quantitative traits varies greatly among different genera due to their different environmental influences and breeding levels [30], so it is generally considered that two SD or two LSD_0.05_ is the minimum level of variation in establishing quantitative trait classification, and there is no feasible method to determine the appropriate level of variation. In this paper, we innovatively used multiple comparisons to classify anthurium, and the results showed that 2.5 times SD or LSD_0.05_ was a suitable interval for anthurium, which will lead to the conclusion that varieties of the same note are not different and varieties with two notes D-value is significantly different, is correct with high confidence. It will reduce the error of distinctness evaluation by note. Anthurium can be propagated by seed or division, but almost all cultivars are now propagated through tissue culture. Compared to other reproductive methods, the morphology of tissue-cultured seedlings is more consistent; our research also showed anthurium varieties have higher uniformity. The lower intra-variety CV means a lower SD, which may be the reason that 2.5 SD was suitable for grading. For other species, especially seed-propagated varieties, 2 SD may be sufficient as a state interval for grading. The sample standard deviation is the average amount of variability in every sample [31]. It tells you, on average, how far each value lies from the sample mean. A high standard deviation means that values are generally far from the mean, while a low standard deviation indicates that values are clustered close to the mean. The t_0.05_ changes with little range while the degree of freedom is less than 3000; the LSD_0.05_ value mainly depends on mean square error (MSE) [12]. MSE is the quotient of the sample standard deviation sum and degree of freedom within the group. While the group number is big enough, the LSD_0.05_ will be similar to SD. The results of this paper confirm this theory. By randomly selecting different varieties for comparison, it was found that when the number of varieties reached 150 or more, there was no significant difference between LSD_0.05_ and SD, while when the number was less than 120, there was a significant difference. Comparing the average SD of different variety numbers, it was found that there was no difference in the average SD between the number of varieties from 20 to 200. However, the results of LSD_0.05_ was different. There is no significant difference between the LSD_0.05_ produced by different variety numbers, but the difference between the maximum and minimum of 10 random selecting on the same variety number increases with the decrease of the number of varieties (analysis results not published). Therefore, it is recommended that the SD method be preferred when the number of varieties is less than 150, and while the variety number is large enough and the characters conform to the normal distribution, LSD or SD methods can be used.

Due to a variety of data and statistical considerations, several dozen MCTs have been developed over the decades, such as Fisher LSD, Tukey’s HSD, Bonferroni, Scheffe, Games–Howell and Newman–Keuls [32]. Among them, Fisher LSD, Tukey and Bonferroni are the most frequently used pairwise comparison tests. Bonferroni is known to be very conservative, while Fisher LSD is sensitive. Even if Fisher recommended using a more stringent alpha while performing his least significant difference procedure (LSD) but researchers find the LSD process inadequate to control a Type I error [33], Tukey’s HSD is probably the most recommended and used procedure for controlling Type I error rate when making multiple pairwise comparisons [34]. Absent linear combinations of means, Tukey’s HSD presents a robust and widely available test for a variety of situations. Due to the large number of varieties to be compared in this experiment and the need to perform pairwise comparisons, Tukey’s HSD method was chosen for multiple comparisons. HSD value also main effect by MSE, as the analytic results of LSD, the MSE also change with the number of varieties. More varieties are to be observed to help increase the stability of MSE. This will affect the result of pairwise comparison. In order to obtain reliable results, when using multiple comparisons for grading, the recommended number of varieties is not less than 50.

Example varieties are provided in the test guidelines to clarify the states of expression of a characteristic [8]. There are many criteria, for example, varieties, such as availability, minimizing the number, and illustration of the range of expression within the variety collection. For quantitative characteristics that need to be observed by measurement, the example varieties should be provided in test guidelines. The main reason why example varieties are used in place of actual measurements is that measurements can be influenced by the environment. By comparing with standard varieties, the same variety in different regions will obtain the same description despite different measurements. In DUS testing practice, each test variety is not directly compared with the example variety; its measurements are transferred to notes according to the grading criteria established by analyzing the measurements of the example varieties. Therefore, when selecting example varieties for measuring quantitative characteristics, experts will choose varieties that represent the average of that state. If multiple comparison methods are used during testing, it is necessary to analyze the tested varieties simultaneously. If the total varieties of a growing trial are less than 50, this method cannot achieve good results.

Multiple comparisons can be analyzed by various software, such as SPSS and GraphPad Prism, but labeling letters requires further manual analysis or other software to achieve. If the amount of data to be compared is relatively large, such as the 251 varieties in this paper, which can form 31,375 variety pairs, it will take a long time to increase the labeling manually. I have written multiple comparison software using Python language, only 6 kb and the package is only 64.1 MB which can be run in Windows system. It only took about 15 min to complete the comparison and labeling of one trait, which was very easy and fast. And the grading criteria can be established by simply classifying the varieties without differences into the same class. The process does not need more adjustments and modifications that rely on experience. This means that with sufficient resources, testers without rich testing experience can accurately transfer the measurement to note, which can greatly reduce the error rate of distinctness evaluation.

This study used anthurium as material to establish the grading criteria by multiple comparisons; the results showed that it was feasible and simple. Other species, especially vegetatively propagated varieties, can benefit from using this method as long as the number of varieties is not less than 50, and whether it is suitable for seed-propagated species needs further verification.

## 4. Materials and Methods

### 4.1. Materials

A total of 251 *Anthurium* varieties seedlings were collected and kept in the Shanghai Sub-center for New Plant Variety Tests, the Ministry of Agriculture and Rural Affairs, China. The seedlings with at least one flower (or flower bud) should meet the quality requirements of commercial seedlings. All plants were planted in pots and placed in glasshouse with shading, cooling, and heating functions. Different size pots were selected according to the type of varieties. Mini varieties were planted in 8 cm pots, moderate varieties in 14 cm pots, and large varieties in 20 cm pots. The Klasmann–Deilmann peat 614 (grain size 15–70 mm, pH 5.5–6.5) was used as substrate. Fertilized alternately with Huaduoduo 23 (15–15–15–7CaO–3MgO) and Huaduoduo 8 (20–10–20 PL) when the plant was in vegetative growth stage and replaced the Huaduoduo 8 by Huaduoduo 1 (20–20–20) when the plants were in flowering stage.

### 4.2. Methods

All the measured quantitative characteristics listed in the national DUS test guidelines for *Anthurium* were observed [35]; they also were listed in UPOV test guidelines for *Anthurium* [36]. The measurement was performed in accordance with the test guidelines: The observation of characteristics was conducted after 3 normal flowers of potted varieties and 1 normal flower of cut flower varieties. All quantitative characteristics were measured in 10 plants, with one sample taken per plant. Leaf blade length and width and petiole length were measured. The leaves corresponding to the largest normal spathe with 1/2–2/3 pistil of the spadix were mature. The peduncle length and thickness, spathe size and spadix length and thickness at the middle were measured in the flower of largest normal spathe with 1/2–2/3 pistil of the spadix mature. All data were obtained in spring or autumn. 

The measurement method is as follows:

Plant size: Measured the height and width of plants with a ruler, plant size was the value of the sum of plant height and width divided by two.

Leaf blade length: Flatten the blade and measure the length from the blade tip to the base using a ruler.

Leaf blade width: Flatten the blade and measure at the widest position using a ruler.

Petiole length: Measured the length from the base of the petiole to the leaf attachment using a ruler. 

Peduncle length: Measured the length from the base of the peduncle to the spathe attachment using a ruler.

Peduncle thickness: Measured the thickness at the middle using a vernier caliper.

Spathe size: Flatten the spathe and measure the length from the spathe tip to the base, recorded as spathe length; measure the broadest position recorded as spathe width using a ruler. Spathe size was the value of the sum of length and width divided by two.

Spadix length: Measured the length from the base to the tip using a ruler.

Spadix thickness at the middle: Measured the thickness at the middle of the spadix using a vernier caliper.

### 4.3. Data Analysis

#### 4.3.1. Describe Statistics and Variability of Quantitative Characteristics

The mean, median and standard deviation (SD) of the quantitative characteristics of each test variety were statistically analyzed using Excel 2019 (version 2305 Buid 16.0.16501.20074) [37].

Subsequently, the maximum, minimum, mean and median of the characteristic among all varieties were found, and the intra-variety coefficient of variation (CV) was defined as the mean of value divided by the mean of SD.

#### 4.3.2. Test for Normality of Quantitative Characteristics

The one-sample K-S (Kolmogorov–Smirnov) test in IBM SPSS Statistics 23.0.0.0 software (IBM corporation, http://b1.go3y.cn/shop/667342.html, accessed on 17 March 2023) [38] was used to analyze whether the collected quantitative characteristic data conformed to normal distribution.

#### 4.3.3. Correlation Analysis of the Quantitative Characteristics

All measurements were transformed into notes according to the grading criteria established by the multiple comparison methods. Then, the data set was used to calculate the correlation coefficients between characteristics using IBM SPSS Statistics 23.0.0.0 software.

#### 4.3.4. Quantitative Characteristic Grading

The state numbers of all quantitative characteristics were required in the national DUS test guidelines for *Anthurium* [35]. 

The 3 methods were used to establish the grade criteria; they were two SD methods, two LSD_0.05_ methods and multiple comparison methods.

The value of LSD_0.05_ was calculated by two-way ANOVA at *p* = 0.05 level. 

In two SD method and two LSD_0.05_ method, the two SD or two LSD_0.05_ was used as interval between states, respectively.

The multiple comparison software was written in Python (version 3.7, https://www.python.org/downloads/, accessed on 17 March 2023), and the pairwise_tukeyhsd method was used to conduct multiple comparisons for each characteristic (statsmodels 0.13.5, https://www.statsmodels.org, accessed on 17 March 2023) [39]. The varieties that did not differ by multiple comparisons were divided into the same state, and the range of each state was determined based on the grouping results, and the corresponding grading criteria were established.

## Figures and Tables

**Figure 1 plants-12-02417-f001:**
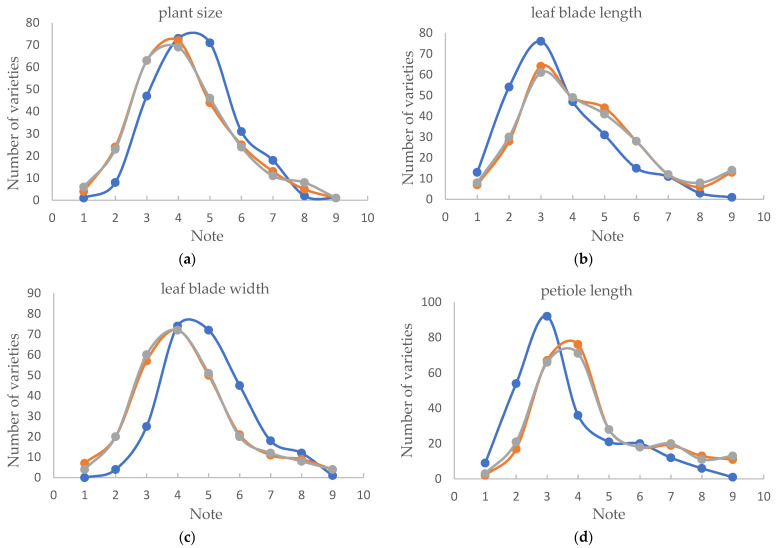
Distribution frequency of different states for 9 characteristics using 3 grading criteria. (**a**): plant size, (**b**): leaf blade length, (**c**): leaf blade width, (**d**): petiole length, (**e**): peduncle length, (**f**): peduncle thickness, (**g**): spathe size, (**h**): spadix length, (**i**): spadix thickness at the middle.

**Table 1 plants-12-02417-t001:** Variations of quantitative characteristics on *Anthurium*.

Characteristic	Minimum	Maximum	Mean	Median	Average SD	Intra-Variety CV
Plant size (cm)	22.95	67.175	39.99	39.34	2.87	7.18
Leaf blade length (cm)	10.36	38.53	19.95	18.84	1.44	7.22
Leaf blade width (cm)	5.25	22.08	11.58	11.09	0.90	7.77
Petiole length (cm)	7.59	49.21	21.86	19.65	2.21	10.11
Peduncle length (cm)	13.40	61.72	31.08	28.97	2.98	9.59
Peduncle thickness (mm)	2.08	6.93	3.91	3.82	0.38	9.72
Spathe size (cm)	4.46	16.29	9.23	9.32	0.83	8.99
Spadix length (cm)	2.13	9.55	4.37	4.17	0.40	9.15
Spadix thickness at the middle (mm)	3.71	10.30	6.32	6.35	0.44	6.96

**Table 2 plants-12-02417-t002:** K-S test of quantitative characteristics of *Anthurium*.

Characteristic	Absolute Extreme Difference	Positive Difference	Negative Difference	K-S Value	Sigma Value
Plant size	0.078	0.078	−0.035	0.078	0.001
Leaf blade length	0.098	0.098	−0.063	0.098	0.000
Leaf blade width	0.081	0.081	−0.043	0.081	0.000
Petiole length	0.152	0.152	−0.082	0.152	0.000
Peduncle length	0.099	0.099	−0.054	0.099	0.000
Peduncle thickness	0.065	0.065	−0.035	0.065	0.013
Spathe size	0.034	0.034	−0.032	0.034	0.200
Spadix length	0.116	0.116	−0.067	0.116	0.000
Spadix thickness at the middle	0.044	0.044	−0.029	0.044	0.200

**Table 3 plants-12-02417-t003:** The correlation coefficient of the nine quantitative characteristics of *Anthurium*.

	Plant Size	Leaf Blade Length	Leaf Blade Width	Petiole Length	Peduncle Length	Peduncle Thickness	Spathe Size	Spadix Length
Plant size	1							
Leaf blade length	0.760 **	1						
Leaf blade width	0.765 **	0.876 **	1					
Petiole length	0.820 **	0.829 **	0.780 **	1				
Peduncle length	0.808 **	0.797 **	0.764 **	0.893 **	1			
Peduncle thickness	0.713 **	0.718 **	0.753 **	0.684 **	0.686 **	1		
Spathe size	0.516 **	0.601 **	0.634 **	0.484 **	0.535 **	0.639 **	1	
Spadix length	0.772 **	0.827 **	0.796 **	0.812 **	0.793 **	0.744 **	0.627 **	1
Spadix thickness at the middle	0.548 **	0.612 **	0.629 **	0.546 **	0.520 **	0.703 **	0.537 **	0.603 **

** Correlation is significant at the 0.01 level (2-tailed).

**Table 4 plants-12-02417-t004:** The state interval established by three methods, respectively.

Characteristic	Multiple Comparison Method	Two SD Method	Two LSD_0.05_ Method
Plant size (cm)	6.60–7.00	5.70	5.60
Leaf blade length (cm)	3.20–3.90	2.90	2.70
Leaf blade width (cm)	2.10–2.50	1.80	1.80
Petiole length (cm)	5.00–5.90	4.40	4.30
Peduncle length (cm)	7.10–7.60	5.90	5.80
Peduncle thickness (mm)	0.86–1.03	0.75	0.70
Spathe size (cm)	2.49–2.60	1.70	2.00
Spadix length (cm)	0.90–1.00	0.80	0.77
Spadix thickness at the middle (mm)	1.00–1.10	0.87	0.84

**Table 5 plants-12-02417-t005:** Grading range of quantitative characteristics of *Anthurium*.

Note	1	2	3	4	5	6	7	8	9
plant size (cm)	19.0	26.0	32.9	39.6	46.4	53.0	60.0	67.0	74.0
	13.5	19.2	25.0	30.7	36.5	42.2	48.0	53.7	59.5
leaf blade length (cm)	13.1	16.3	19.6	23.1	26.6	30.5	34.0	37.5	41.0
	11.6	14.5	17.4	20.3	23.1	26.0	28.9	31.8	38.7
leaf blade width (cm)	4.1	6.2	8.5	10.7	12.9	15.1	17.5	19.6	22.1
	6.6	8.4	10.2	12.0	13.8	15.6	17.4	19.2	22.1
petiole length (cm)	7.1	12.2	17.4	23.0	28.0	33.1	38.3	43.4	49.3
	8.6	13.0	17.4	21.9	26.3	30.7	35.1	39.6	49.3
peduncle length (cm)	9.5	16.5	24.3	31.3	38.3	45.3	53.0	62.0	69.0
	14.1	20.0	26.0	31.9	37.9	43.9	49.8	55.8	61.8
peduncle thickness (mm)	1.5	2.4	3.26	4.12	5.0	5.9	6.93	7.83	8.73
	1.9	2.7	3.4	4.2	4.9	5.7	6.4	7.2	7.9
spathe size (cm)	1.2	3.7	6.2	8.71	11.2	13.7	16.3	18.8	21.3
	5.1	6.8	8.5	10.2	11.8	13.5	15.1	16.8	18.4
spadix length (cm)	1.96	2.86	3.8	4.8	5.7	6.7	7.7	8.7	9.7
	2.2	3.0	3.8	4.6	5.4	6.2	7	7.8	8.6
spadix thickness at the middle (mm)	3.6	4.61	5.61	6.71	7.73	8.8	9.8	10.8	11.9
	4.2	5.0	5.9	6.8	7.7	8.5	9.4	10.3	11.2

Note: The upper row is grading criteria obtained by the multiple comparison method, and the lower row is grading criteria obtained by two SD methods. This table shows the upper state limit of this state. The value of every level is from the upper limit of the lower state to the upper limit of this level.

**Table 6 plants-12-02417-t006:** Distinguished variety pair proportion with different D-value (%).

Variety Pair	With Same Note	With Same Note	Difference of One Note	Difference of One Note	Difference between Two Notes	Difference between Two Notes	Error Rate	Error Rate
Method	The SD Method (Criteria 1)	Multiple Comparison Method (Criteria 2)	The SD Method (Criteria 1)	Multiple Comparison Method (Criteria 2)	The SD Method (Criteria 1)	Multiple Comparison Method (Criteria 2)	The SD Method (Criteria 1)	Multiple Comparison Method (Criteria 2)
Plant size (cm)	2.61	0	52.61	58.41	99.75	100	0.25	0
Leaf blade length (cm)	2.46	0	47.57	60.95	99.20	99.70	0.80	0.30
Leaf blade width (cm)	0.94	0	37.82	51.53	98.34	100	1.66	0
Petiole length (cm)	4.72	0	49.43	61.62	99.66	99.98	0.34	0.02
Peduncle length (cm)	4.37	0	49.60	61.51	99.69	100	0.31	0
Peduncle thickness (mm)	0.63	0	30.31	39.70	98.62	100	1.38	0
Spathe size (cm)	2.97	0	32.60	53.95	81.75	100	18.24	0
Spadix length (cm)	2.03	0	30.55	47.30	98.52	99.98	1.48	0.02
Spadix thickness at the middle (mm)	3.09	0	44.20	54.60	97.24	100	2.76	0
Total	23.84						27.22	0.34

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
