# Peer review of "Grading Criteria of *Anthurium* DUS Quantitative Characteristics by Multiple Comparison"

_plants, 2023, doi:10.3390/plants12132417_

Round 1
Reviewer 1 Report
Discussion should be improved and there is no conclusion or concluding remarks
English is well written, could benefit from a minor review of native english
Author Response
Discussion should be improved and there is no conclusion or concluding remarks.
Response : Thank you for your suggestion. I have revised my discussion and added a conclusion.
Reviewer 2 Report
I have revised the manuscript plants-236671, titled "Grading Criteria of Anthurium DUS quantitative characteristics by multiple comparison". The manuscript attempts to obtain grading criteria for anthurium based on the multiple comparison of different quantitative characteristics. The subject is interesting for breeders of new varieties and to be able to distinguish different varieties, as well as their stability and uniformity of plant characteristics.
I have some comment or concern about this work:
Have the study been carried out in different years, or in different growing conditions, or location? As authors state, the effect of year and site are important in order to establish a grading criteria. The authors should give additional information in Material and Methods.
Table 2: What is the meaning of Absolute extreme difference, Positive difference and Negative difference?. How are they calculated?. Why Absolute extreme difference, Positive difference have the same values?. Authors need to explain these parameter in order to a best undestanding of this table and its values.
Table 3: For table clarity, remove the "**", they are present in all values, just state that all correlation were significant at 0.01 level with a 2-tailed test.
Table 4: Why two interval for the multiple comparison methods?. In this case, how do you select the better interval for grading?
Figure 1: Axis labels are mandatory for a figure. It is clear that x-axis represent the grading levels, but what y-axis represents? percentage, counts,... Authors must provide a better figure with panel letter inside.
General comment:
Although the work is interesting, I have a hard time reading it due to lack of definition of the parameters used, or how they are obtained. Is not clear to me what is the "note" the assign to the varieties, are they the grading levels (1-9)?, why nine interval, which criteria are the authors taken for these levels?. I think a more complete descrition of how this criteria are obtained and used would be necessary for the non-expert reader understand this work.
Author Response
Point 1: Have the study been carried out in different years, or in different growing conditions, or location? As authors state, the effect of year and site are important in order to establish a grading criteria. The authors should give additional information in Material and Methods.
Response 1: The experiment was conducted in multiple years in the same region. Due to all anthuriums are grown in greenhouses, various regulatory measures are taken to minimize the impact of environmental conditions and ensure that they grow in a suitable environment. Therefore, different planting locations were not compared. The observation data is based on the results of the optimal stage for each variety.
Point 2: Table 2:What is the meaning of Absolute extreme difference, Positive difference and Negative difference?. How are they calculated?. Why Absolute extreme difference, Positive difference have the same values?. Authors need to explain these parameter in order to a best undestanding of this table and its values.
Response 2: Absolute extreme difference is the largest difference (in absolute value) between the observed and theoretical cumulative distribution functions. First, query the distribution table to obtain the corresponding theoretical cumulative probability distribution function F (x); Secondly, using the sample data to calculate the cumulative probability of each sample data point, the test cumulative probability distribution function S (x) is obtained; Thirdly, calculate the difference sequence D (x) of F (x) and S (x). Finally, the maximum absolute difference in the difference sequence is calculated, it is Absolute extreme difference. Positive difference means D is positive, that means F (x) > S (x). Negative difference means F (x) < S (x). Because in my test ,the absolute value of Negative difference is litter than positive difference ,so absolute extreme difference is equal to positive difference.
I added explain about absolute extreme difference in 2.2.
Point 3: Table 3: For table clarity, remove the "**", they are present in all values, just state that all correlation were significant at 0.01 level with a 2-tailed test.
Response 3: The correlation signed by “**” is an important result of this analysis, I think it should be reserved.
Point 4: Table 4: Why two interval for the multiple comparison methods?. In this case, how do you select the better interval for grading?
Response 4:In Table 4, the interval for multiple comparisons is shown as an interval, and the intervals between different levels are different, but they are all within the interval shown in the table. For example, the interval of Plant size shown in table is 6.60-7.00, the intervals are 7.00, 6.90, 6.70, 6.80, 6.60, 7.00, 7.00, and 7.00, respectively. To ensure that there is a significant difference between varieties with the two codes different , the maximum of these values can be used as the grading interval.
Point 5: Figure 1: Axis labels are mandatory for a figure. It is clear that x-axis represent the grading levels, but what y-axis represents? percentage, counts,... Authors must provide a better figure with panel letter inside.
Response 5: Thank you for your suggestion. This was my oversight, and the Y-axis represents the number of varieties in that note.
Point 6:Although the work is interesting, I have a hard time reading it due to lack of definition of the parameters used, or how they are obtained. Is not clear to me what is the "note" the assign to the varieties, are they the grading levels (1-9)?, why nine interval, which criteria are the authors taken for these levels?. I think a more complete descrition of how this criteria are obtained and used would be necessary for the non-expert reader understand this work.
Response 6: The "note" refers to the code assigned to the variety according measurement value based on the grading criteria, it also means the grading levels(1-9) . The 9 levels is based on the national DUS test guidelines for Anthurium. I have added explanations in the method.
Reviewer 3 Report
This is an interesting and well written study in purpose to comprehensive evaluation and innovative utilization of germplasm resources, as well as to standardize the description and data of germplasm resources. In my opinion the Introduction section is rather extended and, maybe, should be shorten and be more coherent.
In my opinion the manuscript should be reconsider for publication after major revision. Below are some more specific remarks.
Line 206-213. different font size
Line 237. More details about seedlings (i.e. height, number of leaves etc)
Line 239. pot size, type of greenhouse
Line 255 different font size
Line 250-255. How do you measured the length, width thickness size etc? Please be more analytical. Also the duration of the experiment was not refer in the text. The time for the measurements in each of the 251 varieties (3 normal flowers of potted varie-248 ties and 1 normal flower of cut flower varieties) was about the same, a week or a month as compared each other? Is this a problem for the statistic analysis if the duration is too long?
Minor editing of English language required
Author Response
Point 1: Line 206-213. different font size
Response 1: I'm sorry, I didn't notice the difference in font. I found that the font size is different and has been modified.
Point 2: Line 237. More details about seedlings (i.e. height, number of leaves etc)
Response 2: I added the requirement about seedlings in 4.1.
Point 3: Line 239. pot size, type of greenhouse
Response 3: I added the describe about pot size in 4.1. And the greenhouse was replaced by “glasshouse with shading, cooling, and heating functions”.
Point 4:Line 255 different font size
Response 4: I'm sorry, I didn't found the difference in font.
Point 5: Line 250-255. How do you measured the length, width thickness size etc? Please be more analytical. Also the duration of the experiment was not refer in the text. The time for the measurements in each of the 251 varieties (3 normal flowers of potted varie-248 ties and 1 normal flower of cut flower varieties) was about the same, a week or a month as compared each other? Is this a problem for the statistic analysis if the duration is too long?
Response 5: Because the measurement method of length, width and thickness is explained by illustration in test guidelines, in 4.2, “The measurement were performed in accordance with the test guidelines” was used. I added the describe of method of every characteristics.
Different varieties need different time for normal flowering after seedlings be planted, but the suitable observation period will last a long time. Therefore, this experiment chose spring or autumn with more suitable temperature for data observation. Therefore, the entire experiment lasted for many years, but it is not a problem for statistical analysis.
Round 2
Reviewer 1 Report
Introduction and discussion could be improved and better documented.
As improved, however it would benefit from a further improvement
Author Response
Thank you for your suggestion. I have revised the introduction and discussion.
Reviewer 2 Report
The manuscript has been improved, and can be published.
Author Response
Thank you for your suggestion.
Reviewer 3 Report
The authors have incorporated into the text all the proposed changes and the manuscript shows significant improvement. Therefore it can be published in the journal in its present formAuthor Response
Thank you for your suggestion.